# Perception and Performance of Physical Activity Behavior after Head and Neck Cancer Treatment: Exploration and Integration of Qualitative and Quantitative Findings

**DOI:** 10.3390/ijerph19010287

**Published:** 2021-12-28

**Authors:** Martine J. Sealy, Martijn M. Stuiver, Julie Midtgaard, Cees P. van der Schans, Jan L. N. Roodenburg, Harriët Jager-Wittenaar

**Affiliations:** 1Research Group Healthy Ageing, Allied Health Care and Nursing, Hanze University of Applied Sciences, Petrus Driessenstraat 3, 9714 CA Groningen, The Netherlands; c.p.van.der.schans@pl.hanze.nl (C.P.v.d.S.); ha.jager@pl.hanze.nl (H.J.-W.); 2Department of Oral and Maxillofacial Surgery, University Medical Center Groningen, University of Groningen, Hanzeplein 1, 9700 RB Groningen, The Netherlands; j.l.n.roodenburg@umcg.nl; 3Center for Quality of Life, Department of Head and Neck Surgery and Oncology, Division of Psychosocial Oncology and Epidemiology, Netherlands Cancer Institute, Plesmanlaan 121, 1066 CX Amsterdam, The Netherlands; m.m.stuiver@amsterdamumc.nl; 4Center of Expertise Urban Vitality, Amsterdam University of Applied Sciences, Tafelbergweg 51, 1105 AZ Amsterdam, The Netherlands; 5Mental Health Services in the Capital Region of Denmark, Mental Health Centre Glostrup, University of Copenhagen, Nordstjernevej 41, DK-2600 Glostrup, Denmark; julie.midtgaard.klausen@regionh.dk; 6Department of Clinical Medicine, University of Copenhagen, Blegdamsvej 3, DK-2100 Copenhagen, Denmark; 7Department of Rehabilitation Medicine and Department of Health Psychology Research, University Medical Center Groningen, University of Groningen, Hanzeplein 1, 9700 RB Groningen, The Netherlands

**Keywords:** physical activity, head and neck cancer, mixed methods, perception, cancer survivors

## Abstract

Maintaining or increasing physical activity (PA) may prevent loss of muscle mass and strength after completion of head and neck cancer (HNC) treatment. However, the exercise level of HNC patients may not meet PA guidelines. We aimed to explore HNC survivors’ views on PA, their report of PA, and to compare these with objectively measured PA. Combined qualitative and quantitative data of HNC survivors were explored post-treatment. Data from semi-structured interviews, questionnaires, and objective measurements of PA were collected, analyzed, and integrated. This resulted in the identification of five themes related to prioritizing, day-to-day life, intention, positive feelings, and social support, respectively, in nine HNC survivors (male: n = 5; age: 52–67 years). Objectively measured PA levels were sedentary to low. The lack of intention to increase PA may be related to HNC survivors’ perception that their current activity level is sufficient, despite low levels of measured PA. While some participants feel they need no help with PA, others are insecure about possible harms. Healthcare professionals may be able to help improve PA in HNC survivors with a tailored approach that reduces fear of harm and helps to incorporate higher intensity PA in daily activities.

## 1. Introduction

Head and neck cancer includes cancer of the oral cavity, nasal cavity, paranasal sinuses, oropharynx, hypopharynx, and larynx. More than 90% of all head and neck tumors are squamous cell carcinomas [1]. Due to the fact that these tumors are located in the upper digestive tract, malnutrition is often already a severe problem at the time of diagnosis [2]. Although the majority of patients with head and neck cancer present with locally advanced disease, these patients are generally treated with curative intent. The treatment approach is aggressive and consists of primary surgery or radiotherapy. Surgical resection of high-stage tumors is followed by radiotherapy. This postoperative radiotherapy can be combined with concomitant systemic therapy, such as chemotherapy or biologicals. In patients not eligible for surgery due to an irresectable tumor, combined concomitant chemo-radiotherapy is preferred [3,4]. The morbidity of these treatment modalities often leads to loss of chewing and swallowing function, loss of smell and/or taste, and xerostomia, which in turn may result in inadequate food intake and malnutrition [5,6]. In addition, the disease and treatment can stimulate muscle breakdown by inflammatory activity [7]. Adverse changes in body composition, which are characterized by loss of muscle mass, are often observed in survivors of head and neck cancer (HNC) [8,9]. Low muscle mass is associated with lower survival and higher local-regional cancer recurrence after treatment for HNC [10,11].

Physical activity (PA) can be defined as any bodily movement produced by skeletal muscles that results in energy expenditure [12]. Recent findings suggest that in addition to dietary interventions, maintaining or increasing PA, for instance by incorporating more exercise in routine activities, may prevent loss of muscle mass and strength during and after completion of HNC treatment and may improve quality of life (QoL) [13,14,15]. However, even though exercise may be feasible during and after treatment [13,15,16,17], HNC treatment may result in long-term barriers for PA, including fatigue, problems with swallowing, and pain [18]. As a result, attaining and maintaining an adequate level of PA after treatment may pose a challenge for post-treatment HNC survivors. A survey showed that less than one out of ten HNC survivors reports an exercise level that meets the physical activity guidelines for Americans [19]. In addition, healthcare professionals report that awareness of exercise recommendations and timing is lacking, and as a result, professionals may hesitate to stimulate PA behavior [20].

Multidisciplinary interventions tailored to perception and understanding of PA of HNC survivors could potentially contribute to improved PA behavior and QoL [21]. Current multidisciplinary guidelines for HNC survivors emphasize the importance of health promotion interventions by healthcare professionals [22]. However, development of effective interventions requires insight into HNC survivors’ PA motivation and insight into associations between PA perception and post-treatment PA performance. Motivational aspects of PA behavior can be framed using known behavioral theories, such as the Theory of Planned Behavior (TPB) [23] or the attitude–social influence–self-efficacy (ASE) model [24]. These theories have been used with some success to explain PA behavior in HNC cancer populations, but could not fully explain variation in PA behavior [18,25]. To gain better insight into the psychological mechanisms that can influence PA behavior in HNC survivors recovering from anti-cancer treatment, we conducted an explorative study with qualitative and quantitative aspects and a high level of information density per individual. With this information, we intended to increase our understanding of HNC survivors’ PA behavior and identify possibilities for PA promotion interventions. Thus, the objective of this study was to explore HNC survivors’ views on PA, including their self-perceived PA level, and to compare these with objectively measured PA.

## 2. Materials and Methods

### 2.1. Design

Since the objective of this study was to explore HNC survivors’ views on PA and to compare these findings with objectively measured PA, a mixed methods approach was considered appropriate and the study was reported in accordance with the guideline for Good Reporting of a Mixed Methods Study (GRAMMS) [26]. A two-phase, parallel, mixed data analysis design was applied. Thus, in parallel, but independently, both qualitative and quantitative data were collected and analyzed. The qualitative and quantitative data were combined during the overall interpretation of the results [27]. Concepts from the ASE model served to explore subjective PA behavioral mechanisms on two levels: PA beliefs as described in interviews, and PA level as reported by using questionnaires. The data consisted of a predominantly qualitative analysis of semi-structured interviews framed within the ASE model, descriptive analysis of questionnaires regarding PA behavior, and actual measurements of PA with accelerometers, in a sample of individuals that were treated for HNC at the University Medical Center Groningen, The Netherlands. This study was a sub-study of a prospective observational study on PA in patients with head and neck cancer: the “PA in Patients with head and neck cancer” (PAP) study.

#### 2.1.1. Parent Study

The Medical Ethical Committee of the University Medical Center Groningen (UMCG) approved the parent (PAP) study, under regulation of the Medical Research Involving Human Subjects Act (reference METc 2012/063). We obtained permission to add the semi-structured interviews to the original research protocol for the purpose of the current sub-study. The PAP study was registered in the Netherlands Trial Register, reference NTR4828. The primary outcome of the PAP study was PA as assessed by daily step count and PA level as measured by an accelerometer. Secondary outcomes included self-reported physical performance, self-reported PA level, and PA self-efficacy. The UMCG mainly serves patients from the north-east region of The Netherlands. Dutch-speaking adult patients with a new primary oral or oropharynx tumor to be treated with surgery with curative intent, with or without adjuvant radiotherapy or concomitant chemo-radiotherapy, were considered eligible. Patients with limited mobility or cognitive impairment prior to treatment were not included. Measurements for the PAP study were performed at three time points that were planned consistently with regular hospital visits: one or two days before surgery (T0), six to eight weeks after completion of HNC treatment (T1), and three months after completion of HNC treatment (T2).

#### 2.1.2. Participants

All participants invited to participate in the interviews had to be included in the parent study, i.e., fulfilling the criteria used in the PAP study. The sample for the sub-study was recruited at T1, i.e., six to eight weeks after treatment. The sample of patients that were approached is a convenience sample consisting of all PAP participants who were scheduled for a T1 measurement in the spring of 2013 or in the spring of 2014. The interviews took place after completing the questionnaires and the measurements that were part of the parent study. The interviewers informed the participants about the purpose and nature of the semi-structured interviews by telephone and asked for additional consent prior to the interviews. The participants did not receive the results of the quantitative measurements prior to the interviews, and the interviewers were not present at the quantitative measurements or informed about the measurement results. Nine out of ten patients who were considered eligible agreed to participate. The patient who declined participation reported personal circumstances as the reason. Participants of neither the PAP study nor the current sub-study were financially compensated for their participation.

### 2.2. Qualitative Study Component

The qualitative component of the study aimed to explore the subjects’ experience of PA, with the use of the ASE model as a theoretical framework. 

#### 2.2.1. ASE Model

The ASE behavioral model is an adaptation of the TPB which integrates the concepts of the TPB with the concept of self-efficacy from Social Cognitive Theory [24]. The TPB is based on the assumption that the intention of persons to perform a certain behavior, such as being physically active, can be predicted from their attitude towards the behavior, from their subjective norms, and from their perceived behavioral control [28]. The ASE model is very similar in the assumption that the intention to perform a certain behavior is the central determinant of behavior. However, in the ASE model, perceived behavioral control is adapted into self-efficacy, and subjective norm is adapted into social influence. The adaptation from subjective norm into social influence allows inclusion/integration of role modeling, and social support in addition to social norms [29]. Finally, the ASE model includes the concepts ‘barriers and stimuli’ and ‘knowledge and skills’. In summary, the ASE model suggests that attitude, self-efficacy and social influence lead to intention, which influences behavior. Behavior is also influenced by knowledge and skills and barriers and support [24]. 

#### 2.2.2. Semi-Structured Interviews

Nine semi-structured interviews, guided by the ASE model, took place at the home of the participants or, if preferred, in a quiet room at the UMCG. The interviews lasted 30–50 min. The interviews were performed by two final-year students (one male and one female) who were trained in performing interviews during the bachelor Program of Applied Psychology of the Hanze University of Applied Sciences. The interviewers were trained and supervised by researchers with extensive experience in qualitative research and behavioral research in cancer survivors. Throughout the process, the interviewers reflected on their experiences with the supervisors regularly.

#### 2.2.3. Data Collection 

The interviewers performed the interviews independently and in successive order and used an interview guide based on a topic list related to concepts of the ASE model. 

We had additional interests in the participants’ view of the role of healthcare professionals in promoting PA, since PA recommendations of healthcare providers are associated with higher levels of leisure-time PA in cancer survivors [30]. However, it is also reported that, for instance, not all oncologists promote PA and that referrals to professionals specialized in improving PA are not always provided when needed [31]. Therefore, the topic concerning the role of healthcare professionals was added to the interview guide. All interviews began by asking the participants to describe “What is PA from your point of view?” Sample questions from the interview guide that was used (translated to English language) are presented in Appendix A. 

### 2.3. Quantitative Component

We enriched the qualitative data with quantitative data from the same nine participants, which were obtained in the parent study, and questionnaires were completed before the interviews. The aim was to explore participants’ self-reported PA, self-efficacy and intention, and measurements of objectively measured PA as measured at T1, and to integrate these findings with the findings from the semi-structured interviews. 

### Measures

Age, sex, cancer diagnosis, cancer stage, and type of treatment, i.e., surgery with or without adjuvant radiotherapy or concomitant chemo-radiotherapy, were obtained from the medical records. Weight and height were measured using a platform balance scale (Seca, Hamburg, Germany) and a stadiometer, calibrated to the nearest 0.2 kg and 0.1 cm, respectively. 

Additionally, we included established questionnaires that are based on the key concepts of self-efficacy and intention from both the TPB and the ASE model. The Exercise Self-Efficacy Scale (ESES) [32] and the Stage of Change—Exercise screening instrument (SoC-E) [33] can provide insight into various aspects of lifestyle behavior motives to perform PA. The ESES was used to explore participants’ self-efficacy in performing physical exercise. The score of each item ranges from 0 to 100. A score of 0 indicates that the participant has no confidence in his/her skills of performing physical exercise at all, a score of 50 indicates moderate confidence, and a score of 100 indicates that the participant has highest confidence in his/her skills of performing physical exercise [32]. The SoC-E instrument was used to explore the self-perceived stage of change of participants regarding PA [33]. In the SoC-E, ‘physically active’ is defined as doing activities such as walking, playing sports, bicycling, or dancing for at least 20 min, 3 to 5 times a week. The SoC-E is a single-item scale with a score ranging from 1 to 5, indicating: pre-contemplation, contemplation, preparation, action, and maintenance, respectively. The Exercise Self-Regulation Questionnaire (SRQ-E) was used to explore participants’ self-perceived regulation of regular exercise behavior and is based on the Self-Determination Theory (SDT) [34,35]. SDT differentiates behavioral mechanisms in terms of the degree to which they represent autonomous or self-determined functioning versus less autonomous or controlled functioning. The SRQ-E includes four domains: external regulation, introjected regulation, identified regulation, and intrinsic motivation. All items are stated as assertions and scored on a 1 to 7 Likert scale. The Relative Autonomy Index (RAI) can be calculated from the four domain scores. A predominantly autonomous style will yield a positive RAI score, while a predominantly controlled style will yield a negative RAI score [36].

A SenseWearPro3 (SWP3; BodyMedia, Pittsburgh, PA, USA) accelerometer was worn day and night for three consecutive days, including one weekend day, in the week before or after T1 measurements were planned. Average daily PA level (PAL) was assessed by the average MET value during the measured days, calculated from the minute-by-minute SenseWear data. PAL was classified as follows: PAL 1.00–1.39 = sedentary, PAL 1.40–1.59 = low active, and PAL 1.60–1.89 = active [37]. Average daily time spent on PA with a moderate intensity of at least 3 metabolic equivalents (MET, in minutes), which is considered ‘moderate intensity’, was also calculated for the measured days [38]. A minimum of 21 min of PA at the level of 3 MET per day was used as a threshold for sufficient PA, as a minimum level of 150 min per week of moderate intensive PA in accordance with recent exercises guideline for cancer survivors [39].

### 2.4. Data Analysis

#### 2.4.1. Qualitative Component

The recorded interviews were transcribed verbatim by the interviewers or the project coordinator [40]. The project coordinator randomly listened to some interviews to check the transcripts for correct representation. The transcripts were uploaded in Atlas.ti7 (ATLAS.ti Scientific Software, Berlin, Germany) for text coding. The interviews were analyzed by means of directed content analysis using the seven concepts of the ASE model, i.e., attitude, social influence, self-efficacy, intention, behavior, barriers and motivators, and knowledge and skills, to generate a template for the coding scheme [41,42]. Subsequently, the template was used by the research coordinator to code all text relevant to perception of PA. This process was repeated by a research assistant with a bachelor’s degree in Applied Psychology. The independently coded transcripts from Atlas.ti were compared in harmonization sessions and discussed until consensus for coding was reached. Finally, new overarching themes were allowed to emerge from the codes.

#### 2.4.2. Quantitative Component

Data related to participant characteristics and data resulting from ESES, SoC-E, SRQ-E, and PA measurements were reported in a descriptive manner. Summarized results were reported as median (minimum–maximum). IBM SPSS version 23.0. Armonk, NY: IBM Corp.was used for all quantitative data analyses. 

## 3. Results

### 3.1. Participants

Of the 9 participants, 4 were female. Median age was 65 years (min: 52 years, max: 67 years). Four patients were treated with surgery, three patients were treated with surgery and adjuvant radiotherapy, and two patients were treated with surgery and concomitant chemoradiotherapy. Two patients were diagnosed with tumor class 1, five patients with tumor class 2, one patient with tumor class 3, and one patient with tumor class 4. 

### 3.2. Qualitative Component

Within the 7 concepts of the ASE model that provided the analytical framework, a total of 24 codes were identified. Five overarching themes of PA perception emerged from the codes: (1) barriers and problems prioritizing PA, (2) PA is part of day-to-day life, (3) no need to increase PA (lack of intention), (4) PA is associated with positive feelings or effects, and (5) limited social support and persuasion. The codes and themes are presented with ASE concepts and signature quotes in Appendix A. 

Within theme (1) “Barriers and problems prioritizing”, physical limitations were reported by all participants. These limitations may include limitations due to cancer or cancer treatment such as pain and fatigue. For instance, P34 explained the effort it takes to do day-to-day chores: *“And making my bed, well in the summer it is easier, because I don’t have a heavy duvet blanket on it. But in the winter, I have a woolen duvet, those are very heavy. I really can’t do that”.* In addition, these limitations included pre-existing problems, for instance knee injuries or lung disease. Seven out of nine participants (pp. 28–34) mentioned not being able to undertake additional PA due to external barriers related to prioritizing. These external barriers included work, social obligations, weather conditions, financial problems/constraints, and travel distance. Within theme (2) “PA is part of day-to-day life”, all participants provided low-intensity activity examples of typical PA behavior. When being asked to provide some examples of PA, P31 replied: *“Well, just vacuuming, cleaning the house, clipping something in the garden, mowing the lawn... [further explanation] … It’s not always to do with attitude, you just do things…going upstairs, changing the bed linen … always busy”.* The activities reported by participants included activities in and around the house, volunteer work or informal work, walking or hiking, and riding the bicycle. Two participants (p. 29, p. 32) provided examples of sports activities as an example of their PA behavior. These activities included jogging and gym-based fitness exercise. In theme (3) “No need to increase PA” (lack of intention), eight participants (p. 28, pp. 30–36) reported no interest or an aversion towards incorporating more PA in their daily activities. These participants considered an increase of PA irrelevant, because of the amount of PA at work, because of pre-existing PA habits, or due to age. For instance, P33 stated: *“I will just continue moving the way I do now. And not with something like that… [sports]. I think this is sufficient. […] As long as we keep busy, we’re good, I think. No need to add anything much. I’m also physically active at work.”* On the other hand, eight participants (pp. 28–35) mentioned that PA was associated with positive feelings or effects (theme (4)). P34 provided the following example of the positive effects that PA was associated with: *“This morning I washed the dishes and mopped the kitchen; and just by getting moving, the pain [in my knees] is less bad”.* Finally, within theme (5) “Limited social support and persuasion”, seven participants (p. 28, p. 30, pp. 32–36) reported that they received little or no PA advice from physicians or nurses, or could not recall. Four of these seven participants (p. 30, p. 33, p. 35, p. 36) mentioned a need for help of a physiotherapist or were currently seeing a physiotherapist. The three other participants (p. 28, p. 32, p. 34) stated that they had no need for professional help with PA. Six participants (p. 28, p. 29, pp. 31–34) felt stimulated for PA by their social network. P29 stated: *“But when I’m with colleagues—these people are very much into exercising, and someone will ask you at some point. “What time did you run?” … that can be very stimulating”.* However, five of these six participants (p. 28, pp. 31–34) also mentioned that their social network advised them to be careful. For instance, P32 explained: *“I would have liked to start [working out] a little sooner, but he [points at partner] wouldn’t let me that is why. But otherwise, nothing has changed”.*

### 3.3. Quantitative Component

An overview per participant of characteristics, results of the questionnaires, and objective measurements regarding PA are presented in Table 1. 

#### 3.3.1. Questionnaires

The median overall average ESES score was 68 (min: 45, max: 100; n = 7). The scores in the three domains were mostly balanced, although for most participants, scores for competing demands tended to be slightly higher than scores for internal feelings and situational/interpersonal. The individual scores indicate a moderate to very high confidence in self-efficacy to exercise. The median SoC-E score in these participants was 5 (min: 2, max: 5). Five out of seven participants (p. 28, p. 31, pp. 33–35) reported that they had been physically active for at least 20 min per day, 3 to 5 times a week, for more than 6 months, and thus were in a state of exercise maintenance based on the SoC-E. 

Overall, participants reported a low level of internalized regulation of PA. The regulatory style represented by the subscales was predominantly controlling, and resulted in a negative RAI score for all participants (median −11.3, range −15.8 to −5.0), which indicates a limited level of autonomous behavior with regard to PA.

#### 3.3.2. Objective Measurements

Accelerometer data were available for eight participants. The accelerometer data of the ninth participant was missing as a result of malfunctioning of the device. Median measured PAL/day was 1.2 (min: 1.0, max: 1.6; n = 8). Based on PAL/day, six of eight participants (p. 28, pp. 32–36) were considered mostly sedentary (PAL < 1.4). Five individuals (p. 29, pp. 31–34) met the minimum sufficient level of 21 min of PA at 3 MET intensity.

### 3.4. Integrated Analysis

Combined qualitative and quantitative data at the participant level are presented in Table 2. Four out of five participants (p. 28, p. 31, p. 34, p. 35) that reported “I have been physically active for more than 6 months on a regular basis for at least 2.5 h a week” on the SoC-E questionnaire also reported that they had no intention of increasing their PA level. This confidence in maintaining an adequate level of PA appears to conflict with the fact that in four out of these five participants (p. 28, pp. 33–35), the objectively measured PA level was classified as sedentary. All SRQ-E scores for exercise self-regulation indicate a limited level of autonomous behavior regarding PA. These results appear inconsistent with the ESES scores, in which eight participants (pp. 28–35) reported moderate to high confidence in their skills to perform PA. Overall, self-perceived PA level is higher than actually measured PA. Only one participant (P31) met the recommended guideline for PA [43]. This participant did not report a need for PA improvement and also scored the highest score for self-efficacy and intention. Participants who were committed to PA or motivated for PA and who felt stimulated by their social network tended to have more PA than participants who worried about PA harm.

## 4. Discussion

This study added quantitative data to a qualitative design to explore HNC survivors’ views on PA and their report of PA, and compared these with objectively measured PA. The qualitative exploration led to the identification of five major themes that may contribute to a more nuanced understanding of behavioral patterns in this sample of HNC survivors. While participants reported a relatively high level of self-efficacy and intention in the cognitive theory-based questionnaires, the reported level of autonomous behavior with regard to PA was limited in the self-determination-driven questionnaire. Overall, objectively measured PA was low to mostly sedentary, although five participants met the exercise guidelines for cancer survivors. Participants identified physical limitations, either pre-existing or caused by cancer or its treatment, as an important barrier for PA. Barriers also often appeared to be related to problems with prioritizing PA. Moreover, all participants reported examples of PA characterized by low intensity. Most participants considered daily activities and occupational activities to be synonymous with PA in general. Participants generally saw no need to increase PA, and most participants viewed PA as associated with positive feelings or effects, regardless of their actual PA performance. Finally, social support and persuasion came up as an important theme. While some participants found that they do not need help with PA, others were insecure about possible harms of PA and indicated that they would appreciate help from a physiotherapist. The fear of harm resulting from PA aligns with findings in other studies [18,44].

The combined interview and questionnaire results suggest that the high scores for self-efficacy are congruent with the lack of intention to further increase PA, as reported in the interviews. However, the limited level of autonomous behavior reported in the SDT questionnaire appears to conflict with the high confidence in self-efficacy reported by the participants. On the other hand, this limited level of autonomous behavior seems in agreement with the limited motivation and the lack of intention to increase PA expressed during the interviews. The participants that explicitly expressed a need for professional help with PA in the interviews were among the participants with the least low SRQ-E RAI scores. These participants also scored highest for the SRQ-E domain Intrinsic Motivation. We speculate that these individuals perhaps were aware of their need for support and thought that they could improve their PA level with professional help. 

The apparent contradiction between the high self-perceived self-efficacy and intention and the objective measurements that indicate mainly sedentary PA behavior may be partially explained by the fact that almost all participants described PA as a habit, an automatism, and also something that comes naturally as a result of their occupation. These results may indicate that participants may overestimate their PA performance as a result of unawareness of how much PA is needed and at what intensity to achieve active PA performance and gain the associated health benefits. If PA performance is overestimated, and judged to be at a sufficient level, participants may have little incentive to prioritize PA. 

Our results partially agree with other studies of PA behavior in HNC survivors. A survey on physical activity barriers of exercise behavior in HNC survivors reported that physical symptoms resulting from HNC and its treatment were significantly correlated with PA barriers and low self-efficacy [18]. A more recent survey also found the reported PA barriers were mostly related to cancer or its treatment (i.e., dry mouth or throat, fatigue, shortness of breath, muscle weakness, difficulty swallowing, and shoulder injuries) [45]. Our results also indicate that while physical limitations constitute an important barrier for PA, the physical problems in our participants were more varied and not always related to cancer and its treatment.

### 4.1. Strengths and Limitations

Our study provides unique in-depth insights regarding PA in HNC survivors, due to the design that allowed for exploring and combining data from interviews, questionnaires, and measurements. Moreover, the study included objectively measured PA, while other studies mostly presented self-reported PA [19,46]. However, our study also had some limitations. Firstly, the number of participants in the current study was limited, however, although we focused on patients with primary oral or oropharynx cancer, there was some variance in our sample with regard to treatment and sociodemographic characteristics. Secondly, there were some missing data for the quantitative measurements. On the other hand, we presented a high level of data density for each participant. Participants from our study first performed the quantitative measurements and completed the questionnaires, and were then interviewed. This may have resulted in a higher awareness of PA prior to the interviews. However, the participants did not receive the results of the quantitative measurements prior to the interviews and the interviewers were not present at the quantitative measurements or informed about the measurement results, thus limiting the influence of the earlier measurements on the interviews. Finally, the recruitment for this study was limited to the participants that were included in the parent study. Patients that participated in the parent study may have had a higher than average interest in the subject of PA when compared to the average HNC population. However, if the PA results are better than average in our participants, due to a higher than average interest in PA, PA intention and behavior in a less motivated group of HNC survivors may be even lower. 

### 4.2. Implications for Research and Practice

The analysis in this study was purely descriptive, and qualitative studies applying a more in-depth, interpretative approach are warranted to confirm these findings. However, the combined data in this study should encourage healthcare professionals to improve awareness among HNC survivors that their actual PA performance may be lower than their self-perceived PA performance. Some HNC survivors may have a need for professional guidance to help them prioritize PA, cope with physical limitations, and make them feel safe while exercising. In these instances, healthcare professionals such as oncologists and nurses could play an important role in providing coaching for patients with worries of PA harm, and referral to exercise professionals such as physical therapists could be made for those who need more extensive support, for example those with physical limitations for PA.

Besides PA, head and neck cancer survivors have to deal with other problematic aspects of self-care. For example, food intake may be impaired by loss of smell and/or taste, and impaired chewing and swallowing function. Moreover, oral hygiene is more complex due to changed anatomy. This all needs to be implemented in the daily routine.

Previous research has shown that cancer survivors that participated in an exercise program prioritized exercise more often and lack of interest decreased [47]. Additionally, a tailored approach that reduces fear of harm and emphasizes safety of PA may improve exercise self-efficacy [48]. Finally, helping HNC survivors to understand how they can incorporate more intensive forms of PA in daily activities and work may help them to improve their PA level [49,50].

In survivors that overestimate their current level of PA, a necessary first step may be to improve awareness of their actual PA level, by measuring PA with the use of instruments such as pedometers or accelerometers [51]. For survivors who have an intention to not become more physically active, TPB-based counseling strategies which target intention may provide a suitable approach. However, our results indicate that, for some HNC survivors, PA does not follow from a conscious intention to be physically active. Rather, it is the result of engaging in other meaningful activities (e.g., work). In such instances, using only TPB-based strategies might prove less useful. This impression is similar to the results of a survey of social cognitive correlates of exercise in a large sample of HNC survivors, suggesting that less than a quarter of the variance in intention and only one-sixth of exercise behavior can be explained by the TPB [25]. Therefore, for future research aimed at understanding PA and development of tailored interventions in HNC survivors, we recommend to critically select an appropriate theoretical framework. If not all patients have an explicit intention for PA, it might be valuable to explore the applicability of theories that assume that individuals operate on different levels of motivation and autonomy, such as the Self Determination Theory. The higher scores found in our study for external regulation and introjected regulation could indicate that PA behavior in some patients may be employed if rewards or external approval can be gained [36]. In such instances, it may be helpful if PA is individually monitored, progression is applauded, and benefits of progression are made explicit and match the patient’s interest. 

## 5. Conclusions

Physical limitations, either pre-existing or caused by cancer or its treatment, constitute important barriers for PA in our limited sample of recent HNC survivors. HNC survivors in our sample considered PA to be synonymous with habitual daily activity, which they associated with positive feelings or benefits. The lack of intention to improve PA may be partially explained by the finding that HNC survivors perceive their mostly low-intensity activities as already sufficient, even though the median level of measured PA was sedentary. Finally, although some participants feel that they do not need help with PA, others are insecure about possible harms of PA. Due to the limited sample, larger investigations are needed to confirm how healthcare professionals may be able to help improve PA in HNC survivors with a tailored approach that reduces fear of harm and helps to incorporate higher intensity PA in daily activities. 

## Figures and Tables

**Table 1 ijerph-19-00287-t001:** Questionnaires, and measurements at the participant level.

			Questionnaires					Measures	
ID	ESES ^a^ (*n* = 7)	SoC-E ^b^ (*n* = 8)	SRQ-E ^c^ ER (*n* = 7)	SRQ-E ^c^ IR (*n* = 7)	SRQ-E ^c^ ID (*n* = 7)	SRQ-E ^c^ IM (*n* = 7)	SRQ-E ^c^ RAI ^d^ (*n* = 7)	PAL/day ^e^ (*n* = 8)	Minutes of MET Level 3/day ^f^
P28	NA	5	NA	NA	NA	NA	NA	1.2	20
P29	73	3	7.0	5.5	1.8	3.8	−10.3	1.5	113
P30	NA	NA	NA	NA	NA	NA	NA	NA	NA
P31	68	5	6.3	5.3	1.5	2.0	−12.3	1.6	205
P32	100	4	7.0	7.0	4.8	2.5	−11.3	1.2	83
P33	61	5	6.0	5.8	2.8	3.5	−8.0	1.3	47
P34	70	5	7.0	6.3	1.5	1.5	−15.8	1.0	27
P35	54	5	7.0	7.0	2.5	2.0	−14.5	1.0	8
P36	45	2	5.3	5.8	2.8	4.3	−5.0	1.1	12
n(%)/median (min-max)	**68** **(45–100)**	**5** **(2–5)**	7.0 (5.3–7.0)	5.8 (5.3–7.0)	2.5 (1.5–4.8)	2.5 (1.5–4.3)	**−11.3** **(−15.8–** **−5.0)**	1.2 (1.0–1.6)	37 (8–205)

Summarized scores of multidimensional instruments ESES and SRQ-E are printed in bold print. NA = not available. ^a^ Exercise Self-Efficacy Scale (ESES). ESES = average ESES score. ESES scores range from 0 to 100, 0 = no confidence in skills of performing physical exercise at all, 50 = moderate confidence, 100 = highest confidence in skills of performing physical exercise [19]. ^b^ Exercise Stages of Change (SoC-E): 1 = pre-contemplation, 2 = contemplation, 3 = preparation, 4 = action, 5 = maintenance [20]. ^c^ Exercise Self-Regulation Questionnaire. ER = external regulation, IR = introjected regulation, ID = identified regulation, IM = intrinsic motivation. Scores: 1 = regulation or motivation very untrue for participant, 4 = regulation or motivation somewhat true for participant, 7 = regulation or motivation very true for participant [21,22]. ^d^ Relative Autonomy Index (RAI): 2 × Intrinsic + Identified − Introjected − 2 × External. A predominantly controlled style will yield a negative RAI and a predominantly autonomous style will yield a positive RAI [21,22]. ^e^ PAL/day: PA level/day. PAL 1.00–1.39 = sedentary, PAL 1.40–1.59 = low active, PAL 1.60–1.89 = active [24]. ^f^ Recommended metabolic equivalents minutes of MET 3 medium intensive activity > 150 min/week = >21 min/day [28].

**Table 2 ijerph-19-00287-t002:** Combined qualitative and quantitative results on participant level.

Participant Selected Codes from Interviews	Activity Level Measured by PAL and Minutes Spend in 3 MET PA Per Day	Self-Reported PA Self-Efficacy According to ESES	Self-Reported PA Intention According to SoC-E	Self-Reported Autonomous PA Behavior According to SRQ-E
Fear of harm/inactivity/lack of motivation/lack of support = … At least one or two themes positive for PA = … Reports sports/motivated/feels supported = …	Basal, sedentary or insufficient activity score = … Limited to low activity score = … Somewhat active, active or sufficient activity score = …	Low confidence in PA skills = … (Moderate) Confidence in PA skills = … Highest confidence in PA skills = …	Low PA intention = … (preparing) PA action = … Maintaining PA = …	Little autonomous PA behavior = … Low to moderate autonomous PA behavior = … Predominantly autonomous PA behavior …
P32 Reported PA sports Committed to or motivated for PA Feels stimulated by social network	PAL: sedentary 3 MET minutes sufficient	Highest confidence in skills to perform PA	Action regarding reaching a sufficient level of PA	Little autonomous behavior regarding PA
P31 Reported PA low intensity No intention to improve PA Feels stimulated by social network	PAL: active 3 MET minutes sufficient	Confidence in skills to perform PA	Maintaining a sufficient level of PA	Little autonomous behavior regarding PA
P33 Reported PA low intensity Committed to or motivated for PA Feels stimulated by social network	PAL: sedentary 3 MET minutes sufficient	Confidence in skills to perform PA	Maintaining a sufficient level of PA	Little autonomous behavior regarding PA
P34 Reported low PA intensity No intention to improve PA Committed to or motivated for PA Feels stimulated by social network	PAL: sedentary 3 MET minutes sufficient	Confidence in skills to perform PA	Maintaining a sufficient level of PA	Little autonomous behavior regarding PA
P29 Worries about harm from PA Reported PA sports Committed to or motivated for PA Feels stimulated by social network	PAL: low active 3 MET minutes: sufficient	Confidence in skills to perform PA	Preparing for actions regarding reaching a sufficient level of PA	Little autonomous behavior regarding PA
P28 Worries about harm from PA Reported PA low intensity No intention to improve PA Feels stimulated by social network	PAL: sedentary 3 MET minutes not sufficient	Not Available	Maintaining a sufficient level of PA	Not Available
P35 Worries about harm from PA Reported PA low intensity No intention to improve PA	PAL: sedentary 3 MET minutes not sufficient	Moderate confidence in skills to perform PA	Maintaining a sufficient level of PA	Little autonomous behavior regarding PA
P30 Reported PA low intensity No intention to improve PA	Not Available	Not Available	Not Available	Little autonomous behavior regarding PA
P36 Worries about harm from PA Reported PA low intensity No intention to improve PA	PAL: sedentary 3 MET minutes not sufficient	Lower to moderate confidence in skills to perform PA	Contemplating actions regarding PA	Little autonomous behavior regarding PA

## Data Availability

The data presented in this study are available upon request from the corresponding author. The data are not publicly available due to privacy reasons.

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
