# Peer review of "Perception and Performance of Physical Activity Behavior after Head and Neck Cancer Treatment: Exploration and Integration of Qualitative and Quantitative Findings"

_ijerph, 2021, doi:10.3390/ijerph19010287_

Round 1

Reviewer 1 Report

Your edits have improved the quality of this manuscript and addressed the issues raised in the first review. Please check your manuscript for typos and grammatical errors. Thank you for your work on this interesting research study. 

Reviewer 2 Report

The article improved after revisions. It is in my opinion publishable.

This manuscript is a resubmission of an earlier submission. The following is a list of the peer review reports and author responses from that submission.

Round 1

Reviewer 1 Report

This paper titled “Perception and performance of physical activity behavior after head and neck cancer treatment: Exploration and integration of qualitative and quantitative findings” submitted to International Journal of Environmental Research and Public Health reports findings of a two-phase parallel mixed data analysis reporting head and neck cancer survivors’ views on physical activity (PA) and comparing self-perceived PA level with objectively measured PA. The topic is important and would be of interest to readers of International Journal of Environmental Research and Public Health.

However, I have several points requiring clarification in order to understand the contribution this study makes to the field.

Abstract:

  • The authors stated that “Maintaining or improving physical activity (PA) may prevent loss of muscle mass and strength during head and neck cancer (HNC) treatment” but later stated that “combined qualitative and quantitative data of HNC survivors were explored post treatment”. Please clarify the time period of interest (during treatment or after treatment).
  •  

Introduction:

  • Suggest to provide a clear definition of physical activity in the Introduction as some articles cited were exercise intervention studies. The definition of exercise is different to PA.
  • Please define QoL when it first appears in the text.

Materials and Methods:

  • Suggest to follow a reporting guideline for mixed method research (e.g. Leech NL, Onwuegbuzie AJ. Guidelines for conducting and reporting mixed research in the field of counseling and beyond. Journal of Counseling and Development. 2010;88(1):61-69 or O'Cathain A, Murphy E, Nicholl J. The quality of mixed methods studies in health services research. J Health Serv Res Policy. 2008;13(2):92-98).
  • Data collection: The authors mentioned that they had additional interests in participants’ view of the role of healthcare professionals in promoting PA, please explain why and add the rationale in the Introduction.
  • Suggest to translate the interview guide into English so that the readers can know what questions were included.
  • Please clarify how long the accelerometer was worn?

Results:

  • Given the authors mentioned that they had asked participants about their view of the role of healthcare professionals in promoting PA, what was the result?
  • One participant did not have the accelerometer data, please explain why?
  • In Table 1, it shows that one participant had T class 3 and one had T class 4, but in the text, the authors stated that two patients with tumor class 4. Please clarify.

Discussion:

  • The authors mentioned that participants would appreciate help from a physiotherapist. This was not in the results.
  • In the Implications for research and practice, the authors mentioned that nurses could initially play an important role in providing coaching for patient with worries of PA harm. Why nurses, not other healthcare professionals (e.g. doctors)?
  • Would be good to discuss how different treatments, gender, cancer stage might impact on patients’ perspective on PA. Did the assessment sequence (i.e. PA questionnaires and objective measures before interviews) have an impact on the qualitative results?

Reviewer 2 Report

An interesting study collecting data from patients who survived to head and neck cancer in order to investigate their relationship with physical activity. Although various interviews, questionnaires, and objective measurements have been assessed, the main limitation of the paper is the very low number of participants (just 9) that makes it difficult to draw any statistically significant consideration; 

The introduction must be expanded; a small description of head and neck cancer and its current treatments is necessary, for example: "head and neck cancer is typically a squamous cell carcinoma particularly aggressive that interests this area. Due to its ability to metastasize, the gold-standard treatment is surgery.  However, surgical removal is not always possible for various reasons and in case of relapses, radiation therapy or systemic chemotherapy are proposed" and cite some articles such as: doi: 10.3390/curroncol28040213. and doi: 10.3390/medicina57060563.

Thank You